# Laryngeal, Tracheal, and Bronchial Disease in the Mucopolysaccharidoses: Endoscopic Study

**DOI:** 10.3390/diagnostics10010037

**Published:** 2020-01-10

**Authors:** Paulo Pires de Mello, Anneliese Lopes Barth, Danielle de Araujo Torres, Mariana Pires de Mello Valente, Dafne Dain Gandelman Horovitz

**Affiliations:** Medical Genetics Department, National Institute of Women, Children, and Adolescents Health Fernandes Figueira, Oswaldo Cruz Foundation, Rio de Janeiro 22250-020, Brazil; annebarth@hotmail.com (A.L.B.); danitorres.orl@gmail.com (D.d.A.T.); marianapmello@ig.com.br (M.P.d.M.V.); dafne.horovitz@iff.fiocruz.br (D.D.G.H.)

**Keywords:** mucopolysaccharidosis, laryngeal disease, tracheal disease, bronchial disease, flexible bronchoscopy

## Abstract

Mucopolysaccharidoses (MPS) are genetically determined diseases, leading to a deficiency of enzymes in the glycosaminoglycan (GAG) degradation pathway. The accumulation of GAG occurs in connective tissue in various organs and systems of the body, including the larynx, trachea, and bronchi. Respiratory symptoms are common and severe in these patients, and respiratory disease is a frequent cause of death. A cross-sectional study with flexible bronchoscopy was conducted in 30 MPS patients (6 MPS I, 8 MPS II, 2 MPS III, 3 MPS IV-A, and 11 MPS VI). Only four patients (13.33%) had a normal airway; nine (30%) had mild to moderate disease, 12 (40%) moderate to severe, and five patients (16.67%) had severe disease. Of particular interest, neuronopathic MPS II had the largest proportion of tracheostomized patients who died due to respiratory complications; in MPS IV-A, all patients had significant tracheobronchial deformity with associated tracheomalacia, despite lacking laryngeal involvement. Laryngotracheobronchial disease (LTBD) was associated to longer disease history and was significantly more severe in older patients. Longer use of enzyme replacement therapy did not prevent the progression of LTBD, although the age of therapy introduction may be a crucial factor in lower airway involvement.

## 1. Introduction

The mucopolysaccharidoses (MPS) comprise a group of progressive hereditary diseases originated from the deficiency of glycosaminoglycan (GAG) degrading lysosomal enzymes. The failure in degradation leads to the accumulation of these incompletely catabolized substances in connective tissue throughout the body, especially in the bones, central and peripheral nervous system, liver, blood vessels, skin, cartilage, airways, heart valves, and cornea [1]. Manifestations can be varied and multisystemic, being death before adulthood a common outcome.

Respiratory symptoms are common in this group of patients and may be the first to be described in the clinical history; respiratory disease may also be a relatively frequent cause of death. Airway involvement in MPS can include both upper and lower airways, as well as lead to obstructive pulmonary disease. Glottic and supraglottic thickening, and also abnormalities of the tracheobronchial tree may be found; the trachea can become narrow, tortuous, or obstructed by GAG accumulation. Early recognition of airway involvement in MPS is important and may help to estimate prognosis [1].

The laryngotracheobronchial evaluation of these patients is not uniform, and will be incomplete if only the medical history, physical examination, and chest X-ray are available, especially in preoperative conditions [2,3,4,5]. Thus, a more complete evaluation is necessary, both to quantify the degree of airway involvement due to the natural progression of the disease itself, and also in order to predict the difficulty of airway management in surgical procedures under general anesthesia.

### Laryngeal, Tracheal, and Bronchial Disease in the Mucopolysaccharidoses

Respiratory involvement in MPS includes laryngeal tracheal and bronchial disease. Their progression is not uniform and varies from case to case. Simmons [6], in a comprehensive review of ENT manifestations in MPS, summarized the findings in MPS I, II, III, IV, and VI. Adenotonsillar hypertrophy was universal. In the larynx, enlargement of supraglottic soft tissues, thickening of the ventricular bands and vocal folds were described, as well as deformity of the cricoid and epiglottis cartilages, often leading to respiratory discomfort requiring tracheostomy [7].

The arytenoid mucosa, when flaccid, prolapses into the laryngeal lumen causing stridor and sometimes dysphagia, configuring laryngomalacia. In some patients, the true vocal folds cannot be adequately examined because redundant supraglottic tissue simply does not allow it. False vocal cord hypertrophy may also reduce the laryngeal lumen, being this one of the reasons for obstructive sleep apnea. There is no standardization for the determination of laryngeal mucosa changes in rigid or flexible endoscopy performed in these patients. The Cormack–Lehane classification, generally used to describe the larynx in intubation, was considered insufficient, as it only describes the view of the glottic space during direct laryngoscopy [8].

Intrathoracic obstruction is also a common complication, with lower airway involvement primarily caused by tracheomalacia and reduction of the tracheal lumen that progresses over time secondary to submucosal and cartilaginous infiltration by GAG. The trachea of these patients may be narrow, tortuous, or obstructed by soft tissues. It may have its wall thickened by fibrosis, inflammation, and granulation tissue. Depending on the site and severity of the obstruction, patients may present with stridor, dyspnea, cyanosis, dysphagia, and cough [7,9].

In some patients, tracheal narrowing is attributed to complications of orotracheal intubation or tracheostomy; bronchoscopy may reveal excessive formation of granulation tissue projecting into the lumen of the cannula at the expiratory phase of the respiratory cycle. Laryngeal, tracheal, and bronchial evaluation is of paramount importance in patients with MPS, both to diagnose the extension of disease involvement and also before interventions which require general anesthesia. Flexible endoscopy is an effective method to diagnose such involvement both preoperatively and postoperatively (very important as an extubation aid) [10].

The trachea can take different shapes in patients with MPS. In the D-shape, the transverse diameter is larger than the anteroposterior diameter by its collapse in this last direction; W-shape, where the trachea becomes elliptical with a large anteroposterior diameter larger than the transverse; O-shaped by the presence of slight (posterior) membranous thickening; and finally the C-shaped with similar anteroposterior and transverse diameters. If the diameter and shape of the trachea can be assessed prior to orotracheal intubation, local injury will be minor or possibly avoided, with significant reduction in perioperative and postoperative complications, especially by choosing a smaller orotracheal tube than usual [10]. The severity of respiratory impairment varies according to the MPS subtype. According to Yeung et al. [7], clinically significant airway obstruction was found in 70% of patients, being tracheostomy required in 11%. For Soni Jaswiel et al. [11], more than 16% of MPS children will require tracheostomy, either as a consequence of complete airway obstruction or extubation failure following general anesthesia. In these patients, tracheostomy is associated with significant morbidity, which may in part be attributed to tracheobronchial abnormalities (narrow trachea, short neck, involvement of the cervical spine and difficulty in neck extension).

The assessment of laryngeal, tracheal and bronchial disease should start with the clinical history and physical examination. Chest X-ray may suggest a change in tracheal shape when its transverse diameter is reduced, but a reduction in the anteroposterior diameter cannot be detected on conventional radiography. Reconstructed chest tomography may provide valuable information on the possible deformities in the trachea and bronchi, as well as allow the measurement of transverse and anteroposterior diameters at these sites. However, tomography without dynamic study (forced inspiration and forced expiration) does not inform about possible tracheomalacia present in these patients; in addition, there may be underestimation of the craniocaudal extension of a tracheal stenosis. Thus, flexible endoscopy is the exam that provides the most information about the disease in these topographies; it documents airway deposits, evaluates the larynx dynamics in its respiratory, phonatory and sphincter functions, detects tracheobronchial deformities and is fundamental in the assessment of tracheobronchomalacia and its association with the respiratory cycle dynamics.

A characteristic of patients with MPS is the difficulty of submitting them to anesthetic procedures for surgery; laryngoscopy and intubation in these patients are often difficult due to the characteristics of the disease itself. However, there is often the need to perform surgical procedures in these patients to correct the systemic manifestations of the disease.

Anesthetic complications occurred during surgical procedures are associated with airway obstruction, with subsequent ventilation and oxygenation difficulties. Laryngoscopy and intubation are often difficult in these patients due to the presence of craniofacial abnormalities, short neck, stiffness of the temporomandibular joint, macroglossia, deposits in the larynx in addition to trachea deformity. In addition, some patients present instability of the cervical spine, being recommended to avoid hyperextension of the neck, which undoubtedly hinders laryngoscopy and intubation; neck movements during intubation are associated with the risk of spinal cord compression and consequent quadriparesis. It is recommended to ensure spontaneous ventilation until intubation in these patients, because bleeding from some abnormal airway deposit with their collapse may be harmful to the patient.

The presence of a difficult airway specialist who has familiarity with intubation difficulties and facilities in the use of broncofibroscope is advisable. Spontaneous breathing induction with a volatile agent, the use of laryngeal mask and broncofibroscope are recommended to guide the most difficult intubations. Flexible bronchoscopy anticipates these difficulties and informs anesthesiosolgists the degree of difficulty to be found, besides being an effective method of intubation in these patients.

## 2. Materials and Methods

A cross-sectional study was conducted in patients diagnosed with MPS, who underwent complete endoscopic examination of the larynx, trachea, and bronchi (flexible bronchoscopy), that were scheduled an exploration to assess the disease involvement, and patients who had not previously undergone intubation. Patients who were admitted due to clinical or anesthetic complications did not complete the proposed exam, and patients who were transplanted before six months of age were excluded from the analysis.

All exams were performed at the operating room. Initially, flexible nasopharyngolaryngoscopy was performed to evaluate laryngeal deformities, laryngomalacia, redundancy of supraglottic tissues, vocal fold mobility, and local GAG infiltration. After laryngeal examination under local anesthesia, flexible bronchoscopy was performed under general anesthesia, maintained by laryngeal mask with sevoflurane and spontaneous ventilation. The Olympus MF 160 3.5 mm videobronchoscope was inserted through the laryngeal mask and the presence or absence of tracheal deformity was assessed, as well as the type of deformity, lumen obstruction due to probable GAG deposition, evaluation of transverse and anteroposterior diameters, presence of tracheomalacia and its variation on spontaneous breathing, as well as the main bronchi were observed.

Due to the lack of uniformity of laryngotracheobronchial disease (LTBD) in the literature, a classification was created for better evaluation and comparison of findings. Laryngeal disease (LD) was classified as absent, mild, moderate, and severe according to the observed changes and points were attributed (0, 1, 2, and 3 respectively). The same occurred with tracheobronchial disease (TBD), which was also classified as absent, mild, moderate, and severe, to which points were also assigned (0, 1, 2, and 3 respectively). All videos/images from the exams were reviewed and classified independently by three experienced endoscopists from our group, and the mean score among these evaluations was used as the result for each variable in each patient. The LD and TBD scores were then added and a final score was obtained for LTBD, which was classified as absent (0), mild to moderate (1 and 2 points), moderate to severe (3 and 4 points), and severe (5 and 6 points) (Figure 1).

Informed consent was obtained for all individuals. The study (registered under the number CAAE: 0074.0.008.000-11) was approved in 12 December 2011 by the local ethics committee on human research (Comitê de Ética em Pesquisa-Instituto Fernandes Figueira) and followed the ethical principles of the Declaration of Helsinki.

A descriptive analysis of the clinical characteristics of the study population was performed, followed by a descriptive analysis of LTBD in the general group and then divided by MPS subtypes. A comparison between MPS subtypes in regards to the respiratory endoscopic disease was also performed. Finally, an analysis of the association between LTBD and length of disease, duration of enzyme replacement therapy and age of treatment initiation was undertaken. For the latter analyses, Fisher’s exact test was used, as it is more adequate when the sample size is small.

## 3. Results

Thirty-two patients underwent endoscopic airway examination and two of them were excluded from the analysis, both belonging to the MPS II group, both with prenatal diagnosis of MPS and undergoing very early hematopoietic stem cell transplantation (before age of three months). Thus, the remaining 30 patients were divided as follows: 6 MPS I, 8 MPS II, 2 MPS III, 3 MPS IV, and 11 MPS VI. The group consisted of 21 male (70%) and 9 female (30%) patients, with an average age of 11.10 years (1 to 39 years), 25.43 kg of weight (10 to 55 kg), 111.13 cm in height (84 to 152 cm), and 4.14 years of age at diagnosis (3 months to 32 years).

From a clinical point of view (without access to polysomnographic data), 80% reported snoring and 53.34% sleep apnea. In contrast, only 16.67% had some type of stridor and 26.67% some degree of dyspnea. Cyanosis episodes were reported in 9 patients (30%), two of them related to enzyme infusion and the remaining seven related to respiratory symptoms (Figure 2). Regarding surgical procedures before the diagnosis of the disease, 23.33% of the patients required some surgical intervention before MPS was diagnosed.

Of the 30 patients, 24 (80%) underwent enzyme replacement therapy (ERT), with a mean time of treatment of 6.43 years (4 MPS I, 8 MPS II, 3 MPS IV-A, and 9 MPS VI; variance of treatment: 2 to 12 years). One MPS I patient underwent transplantation at 8 months and is not under enzyme replacement. Of the remaining five patients, two have MPS III, for which there is no enzyme therapy, while the other three had recently been diagnosed and were not yet on specific therapy (one MPS I and two MPS VI) (Table 1).

Endoscopically, 36.67% had some degree of laryngeal deformity, all in the supraglottis (arytenoid edema, false vocal fold thickening, and post-cricoid edema), sparing the vocal folds and subglottis. Supraglottic tissue redundancy occurred in 63.33% of patients and epiglottis thickening, sometimes without cartilage support, occurred in 56.67% (Figure 3). Laryngomalacia was observed in 10 patients (33.33%). Four of our patients underwent tracheostomy (13.33%), one being an MPS I and the other three MPS II. The only urgent tracheostomy was performed on the MPS I patient during a complication of elective surgery (inguinal hernia repair and carpal tunnel syndrome release), with difficulty in intubation during anesthetic induction. The remaining tracheostomies were electively performed due to advanced laryngeal disease (all patients with neuronopathic MPS II).

Diffuse or segmental tracheal deformity, characterized by alteration of the anteroposterior or transverse diameter, with consequent associated tracheomalacia, was present in more than half of our patients (53.33%). The main bronchi, on the other hand, only presented some type of deformity in 13.33% of the patients (Figure 4). Endotracheal and endobronchial obstruction due to probable GAG deposition occurred in four of the examined patients (13.33%), causing death in two, both with the neuronopathic form of MPS II, (Figure 5).

After reassessment of all examinations by three endoscopists in our group, the patients were classified according to LTBD (sum of laryngeal disease and tracheobronchial disease): four patients (13.33%) did not present any abnormality in the endoscopic exam (all with MPS VI); nine patients (30%) had mild to moderate disease; 12 patients (40%) had moderate to severe disease; and finally five patients (16.67%) had severe disease (three with neuronopathic MPS II and 2 with MPS VI) (Table 2).

In the association between LTBD and length of disease, measured by the patient’s age in years, respiratory disease was significantly more severe in older patients (*p* = 0.003). Age was divided into three categories (up to five years, between six and 12 years, and over 12 years); approximately 93.3% of patients older than 12 years had moderate to severe or severe disease while 100% of patients younger than five years had mild to moderate or absent disease.

Time under specific treatment (enzyme replacement therapy) was divided into three categories: less than five years of treatment, between five and nine years, and over nine years; 66.7% of patients under five years of treatment had absent or mild to moderate disease, while between five and nine years, 78.6% had moderate to severe or severe disease, and 100% of patients treated for more than nine years had moderate to severe or severe disease. Although no significancy was confirmed by Fisher exact test (*p* = 0.224), data seem to demonstrate that prolonged treatment did not prevent disease progression.

The same trend was observed when evaluating age at the beginning of ERT and LTBD. Patients who started treatment earlier had less severe endoscopic disease; Almost half of the patients (44.4%) who started treatment by the age of three years have mild to moderate or absent disease, while 75% of patients who started treatment between four and seven years have moderate to severe disease, and 100% of patients who started treatment after age seven years have moderate to severe or severe disease.

## 4. Discussion

The lower airways are common sites of GAG deposits, resulting in distortion of the anatomy and physiology of the respiratory tract in patients with MPS. We observed a clear worsening of LTBD in older patients; 93.3% of patients older than 12 years had moderate to severe or severe LTBD. In contrast, all patients under five years of age had mild to moderate or absent disease at endoscopic examination.

Specific airway involvement may manifest through supraglottic and glottic thickening or diffuse tracheobronchial involvement by tracheobronchomalacia, endoluminal obstruction by GAG, and/or mediastinal compression by these same deposits. Eighty percent of our patients reported snoring and 53.34% sleep apnea. In contrast, only 16.67% had some type of stridor, 26.67% some degree of respiratory distress, and 30% episodes of cyanosis, which may suggest some adaptation mechanism to the progressive worsening of the respiratory disease observed in this group of patients.

The progressive respiratory involvement is caused by disease at multiple levels: macroglossia, diffuse laryngeal mucosal thickening, extrinsic compression of trachea by GAG deposits, and tracheobronchomalacia. In neurologically impaired patients, swallowing disorders and repeated bronchoaspiration may aggravate breathing problems. Respiratory infections are frequent and should be treated aggressively, as they increase the risk of airway obstruction.

Patients who started treatment earlier had less severe LTBD than those who started later; 44.4% of patients who started treatment before age three years have mild to moderate or absent disease, 75% of those who began treatment between four and seven years have moderate to severe or severe disease, and ultimately all patients who started treatment after age seven years have moderate to severe or severe LTBD.

There is still no universal protocol for classifying abnormalities in the laryngeal mucosa observed in rigid or flexible endoscopy performed in MPS patients. The Cormack–Lehane classification was insufficient for these patients, as it only describes the visualization of the glottic space [8]. The use of a standardized classification to describe such changes could be useful for properly assessing the risk of intubation and extubation in MPS, as well as assisting in information on the natural course of the disease and the effectiveness of ERT. The most severe alterations in the mucosa of the larynx were found in the posterior larynx, arytenoids, and false vocal folds, as described by Keilman et al., and were more common in MPSII [12]. Walker et al. described 34 children who underwent general anesthesia for 110 procedures. The greatest difficulty in intubation was found in MPS II patients, and this was attributed to mucosal changes found in these patients’ larynx [13].

Morimoto et al. [14] performed endoscopic examination of 35 patients during spontaneous ventilation, also proposing their own classification. When absolutely normal the larynx was classified as grade zero; Grade 1 classified patients with epiglottis and remaining supraglottic edema, without excessive mucosal redundancy, without inspiratory obstruction, and without obstruction to endoscopic evaluation of the true vocal folds by the false vocal fold hypertrophy. Finally, in grade 2 were patients with excessive supraglottic mucosal redundancy, inspiratory obstruction, and significant false fold hypertrophy preventing proper examination of the true vocal folds. He reported 23 patients (66%) with deformity of the laryngeal architecture by GAG deposits [14].

Endoscopically, 36.67% of our patients had some degree of laryngeal deformity, all in the supraglottis (arytenoid edema, false vocal fold thickening, and post-cricoid edema), sparing the vocal folds and subglottis. Supraglottic tissue redundancy occurred in 63.33% of patients and epiglottis thickening, sometimes without cartilage support, occurred in 56.67% of patients. The classification proposed by our group divides laryngeal disease into absent, mild, moderate and severe; laryngeal disease was absent only in eight patients (26.67%). The remaining 22 patients (73.33%) had some degree of laryngeal disease already in place (six severe, two moderate, and 14 mild).

As respiratory disease progresses, tracheostomy may be required as a treatment for respiratory obstruction caused by advanced laryngeal disease or as an urgency in anesthetic procedures (intubation and extubation difficulty). In a 1992 study, 44% of children with MPS who underwent adenotonsillectomy due to airway obstruction required tracheostomy [15]. This procedure may be associated with significant postoperative morbidity and mortality, partly attributed to tracheobronchial abnormalities. Of our 30 patients, four (13.33%) were tracheostomized, one MPS I and the other three MPS II. Only in one of them, the patient with MPS I, it was an emergency procedure required during an elective surgery, as intubation in anesthetic induction was impossible. The other three, with neuronopathic MPS II, underwent elective tracheostomy for advanced laryngeal disease; two among these died due to disseminated tracheobronchial disease; the other has advanced tracheal and bronchial disease and is using an adjustable tracheostomy cannula to overcome proximal obstruction due to GAG deposits.

In patients with MPS, tracheostomy is associated with significant morbidity and mortality, which can partly be attributed to tracheobronchial abnormalities. It can be technically difficult to be performed by the anatomical characteristics of these patients (narrow trachea, retrosternal, short neck, and involvement of the cervical spine). Such anatomical characteristics lead to an increase in accidental decanulization rates and associated mortality. Obstruction of tracheostomy cannula is common due to excess secretion, typical in these patients. Repositioning or cannula exchange is always very difficult technically in patients with the anatomical peculiarities of this group. Broncofibroscopy may be necessary to ensure a patent airway, as it is through it that the proper position of tracheostomy cannula and tracheobronchial patency is confirmed.

While in many organs and systems affected by MPS the GAG accumulation leads to reduced mobility (heart valves, joints, and skin), the trachea, on the other hand, is affected by weakening of the supporting cartilage, resulting in airway collapse, possibly also due to associated inflammation. In an animal model of MPS VI, in which tracheal collapse occurred at 9 months of age, two forms of anti-inflammatory resulted in increased tracheal lumen and strengthening of its wall [16,17].

Tracheal involvement in the MPS may present as tracheomalacia, resulting from tracheal deformity and weakening of the supporting cartilage, mediastinal compression by GAG deposits and endoluminal obstruction also by deposit aggravated by associated inflammatory reaction. Although tracheomalacia initially refers to the weakening of the supporting cartilage, it may also be characterized by posterior membrane thickening and reduced tracheal diameter. Of 35 patients studied by Morimoto et al. [14] by chest tomography, 17 had abnormal trachea with collapse in the transverse and/or anteroposterior direction during the respiratory cycle. More than half of our patients (53.33%) had tracheal deformity with abnormal ratio of transverse and anteroposterior diameters of the organ at endoscopic evaluation, which was not accompanied by the main bronchi, altered only in 13.33% of patients. Tracheal deformity was always associated with lack of cartilage support and collapsed organ lumen during the respiratory cycle, and varying degrees of tracheomalacia. It should be emphasized that MPS IV-A patients had moderate to severe LTBD caused exclusively by major tracheal deformity without laryngeal disease.

Endotracheal and endobronchial obstruction due to probable GAG deposition occurred only in four patients (13.33%), three MPS II (all neuronopathic) and one MPS VI. Two of these MPS II died; both had aggressive progression of tracheobronchial disease, despite using adjustable tracheostomy cannulae to overcome the areas of obstruction caused by these deposits. Considering two of the possible major pathophysiological mechanisms of tracheal involvement in MPS (tracheal deformity with secondary tracheomalacia and endoluminal obstruction by deposits), 66.66% of our patients had some form of tracheal disease, with a marked predominance of tracheal deformity.

MPS patients may develop respiratory tract mucosa complications more often than other patients when external synthetic materials such as endotracheal tubes or tracheostomy cannulae come into contact with the organ lumen. Granulomatous changes with lymphocyte infiltration may occur in the respiratory epithelium by mechanical stimulation of the tracheostomy tube itself and especially by GAG deposits. Finally, progressive obstruction due to recurrent infections and pneumonia may result in severe hypoxemia and acute respiratory failure. Two patients in the study group (neuronopathic MPS II) presented this type of progression: despite tracheostomy, the tracheobronchial disease progressed widely and both died from respiratory complications. GAG deposits in these patients were more pronounced in the area of trauma by the cannula tip.

Treatment of airway obstruction in MPS has been moderately successful. GAG accumulation in tonsils and adenoids with consequent hypertrophy makes adenotonsillectomy a very common surgery in this patient group. The relief achieved by surgery is limited, which can be understood by the multifactorial pathophysiology of airway obstruction.

Despite the treatment options for MPS, airway involvement remains significant in several subtypes. Current therapies can slow the progression of airway disease, but not completely stop it. ERT is used in types I, II, IV-A, VI, and more recently in type VII, but significant respiratory problems persist, although pulmonary function may improve [18,19]. Harmatz et al. conducted an ERT study in MPS VI and found a reduction in sleep apnea in non-tracheostomized patients [20]. Of the 15 patients longitudinally followed by Keilmann et al. before and after ERT, there was no uniform pattern in laryngeal mucosa behavior. Some MPS II patients endoscopically reassessed six months after initiation of treatment had worsened. Such finding was explained by the author to be caused by partial depletion of GAG deposits with consequent mucosal redundancy and prolapse into the laryngeal lumen on inspiration [12].

Of the 30 patients in our sample, 24 (80%) received ERT for a mean 6.43 years (variance 2 to 12 years). Almost half of them, despite being treated, presented moderate to severe LTBD (12 patients). Only two patients had normal examination (8.32%), while five patients (20.84%) had mild to moderate disease, and five severe disease (20.84%). Although we are not facing a longitudinal study, where endoscopic examinations could be compared over time, it is possible that prolonged ERT did not prevent disease progression. The longer the treatment, which also means the longer disease time, the more severe LTBD was.

One MPS I patient from our sample underwent hematopoietic stem cell transplantation at eight months, after diagnosis at three months. This child already has moderate segmental tracheal deformity (cervical trachea), although not symptomatic from the respiratory point of view. Only a rigorous clinical and endoscopic follow-up will be able to evaluate the progression of LTBD in this patient.

Belani et al. [21] examined 30 anesthetized patients for various procedures; of these 30, 15 had airway obstruction and 28 had been transplanted. All patients who had noisy breathing, a history of obstructive sleep apnea or desaturation before transplantation showed a significant reduction in their symptoms. Oropharynx size, tongue appearance and laryngeal visualization have all improved considerably.

Andrea et al. [7] reported 14 transplanted patients; 11 of the 13 who had obstructive respiratory symptoms and underwent pulmonary function testing prior to transplantation showed improvement in these symptoms based on new function tests after transplantation. Transplantation and other new therapeutic approaches have not, however, been consistently studied in their effects on laryngeal and tracheobronchial disease.

It is worth of note that one of the patients examined by our team and not included in sample, as he was transplanted at two months, a MPS II boy with predicted neuronopathic variant by family history, had normal airways when endoscopically examined.

## 5. Conclusions

Respiratory disease and its sometimes fatal progression is well recognized literature. Flexible bronchoscopy proved to be a fundamental exam in understanding the pathophysiology of LTBD, being safe when performed in an appropriate environment and by trained professionals.

Only four patients from our sample had no endoscopic changes (13.33%); the remaining 26 patients had some degree of LTBD. MPS II in its neuronopathic form presented the least favorable evolution from the respiratory point of view, including the largest proportion of tracheostomized patients who died due to respiratory complications. It is also worth mentioning the behavior of LTBD in MPS IV-A, since none of the patients had laryngeal involvement, but in contrast, all had significant tracheobronchial deformity with associated tracheomalacia.

When associating the lenght of disease progression with LTBD, it was observed that respiratory disease is significantly more severe in older patients, confirming its progressive character. Prolonged enzyme replacement therapy did not prevent disease progression, however, age at initiation of treatment may be a crucial factor in LTBD, as those who had later introduction of treatment were worse compared to patients who started treatment earlier.

The rarity of MPS compels us to exercise caution in our conclusions and, above all, to avoid generalizations. Particular issues to be explored may deserve further studies with longitudinal follow-up.

## Figures and Tables

**Figure 1 diagnostics-10-00037-f001:**
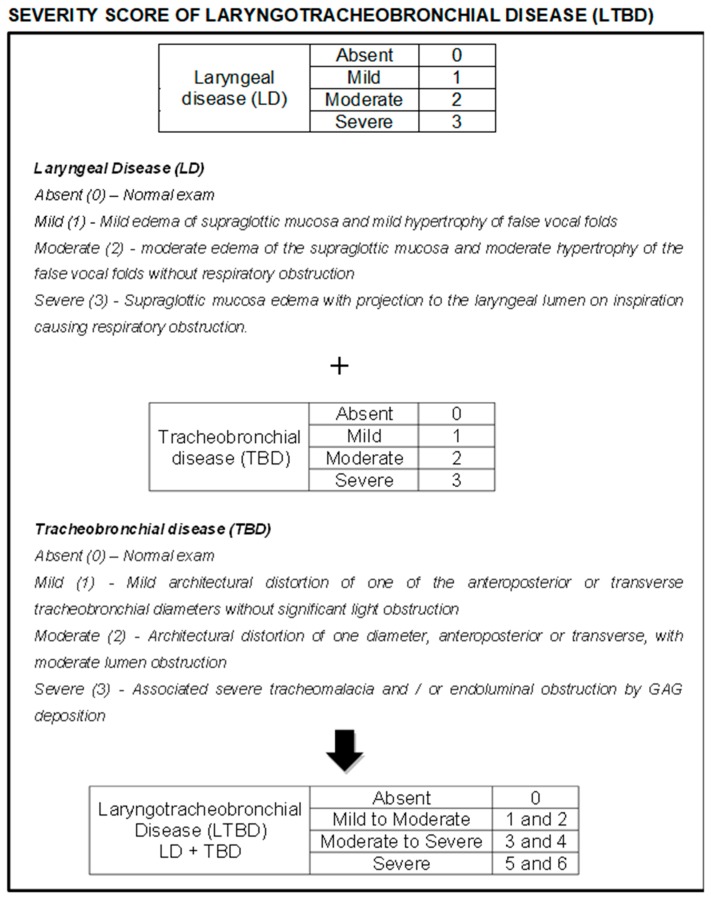
Classification of laryngotracheobronchial disease in mucopolysaccharidoses (MPS).

**Figure 2 diagnostics-10-00037-f002:**
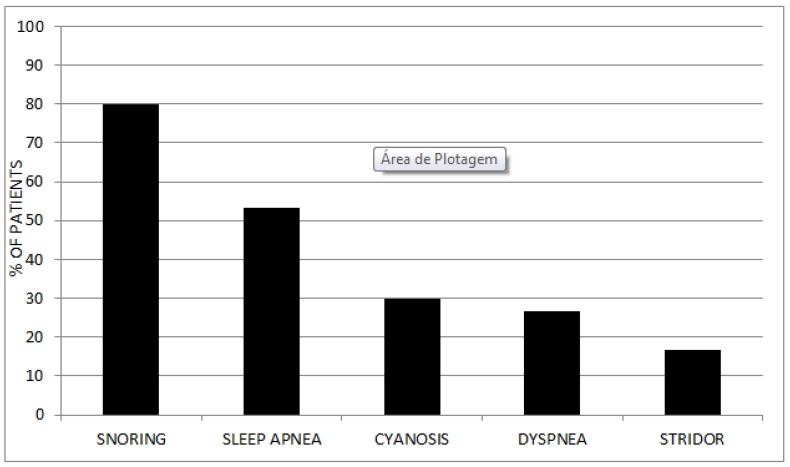
Percentage of clinical findings (as referred by patient history) in the mucopolysaccharidoses (MPS) patients evaluated.

**Figure 3 diagnostics-10-00037-f003:**
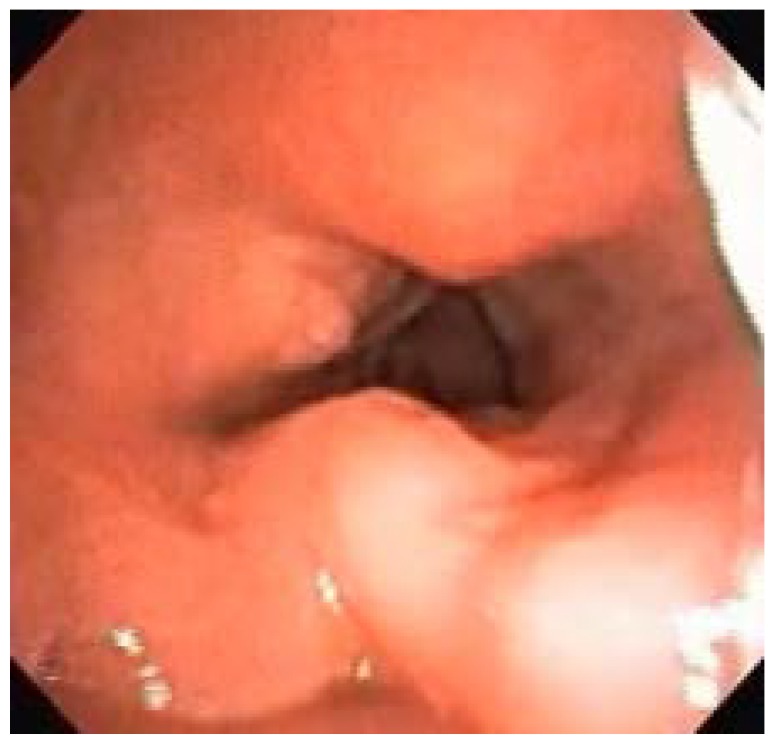
MPS I patient with significant supraglottic involvement (infiltration of false vocal fold, posterior larynx, and epiglottis).

**Figure 4 diagnostics-10-00037-f004:**
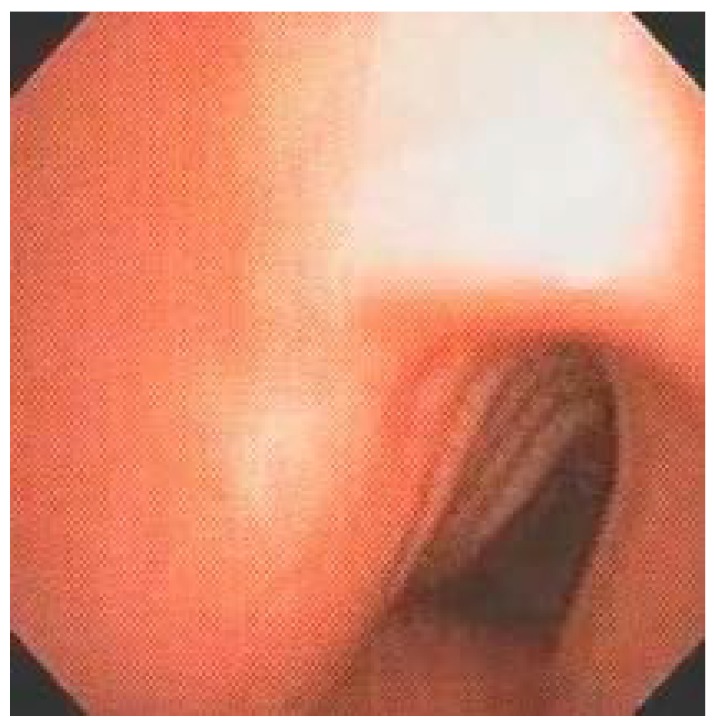
Proximal tracheal deformity in a MPS IV-A patient from the sample.

**Figure 5 diagnostics-10-00037-f005:**
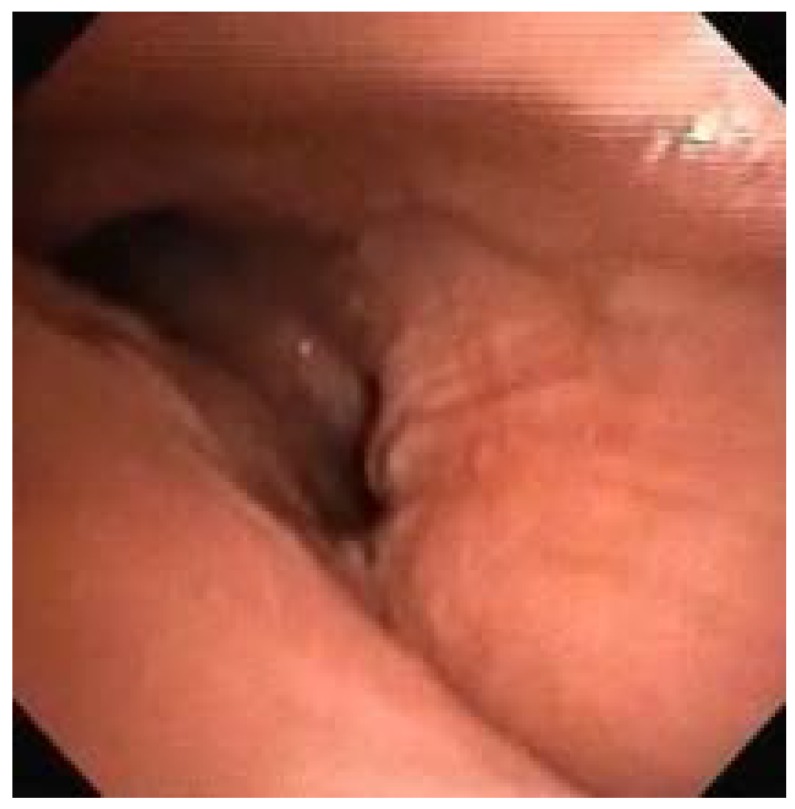
Significant glycosaminoglycan (GAG) infiltration in a MPS II patient (infiltration and architectural distortion of the distal trachea and right main bronchus).

**Table 1 diagnostics-10-00037-t001:** Clinical characteristics of 30 MPS patients.

No	MPS Type	Gender	Age (Years)	Weight (Kg)	Height (Cm)	Age at Diagnoses (Years)	Treatment	Age AT Initial Ert (Years)	Ltdb
1	I	M	22	31	131	11	ERT—11 YEARS	11	Moderate to Severe
2	I	F	4	17	87	0.4	ERT—3–8 MONTHS HSCT—8 MONTHS	8 MONTHS	Mild to Moderate
3	I	M	5	13	96	2	ERT—3 YEARS	2	Mild to Moderate
4	I	M	1	10	84	1	-	-	Mild to Moderate
5	I	M	6	18	102	1.5	ERT—5 YEARS	1	Moderate to Severe
6	I	F	14	32	129	2	ERT—13 YEARS	1	Moderate to Severe
7	II	M	8	23	117	3	ERT—4 YEARS	4	Mild to Moderate
8	II	M	5	26	115	2	ERT—3 YEARS	2	Mild to Moderate
9	II	M	39	55	152	32	ERT—6 YEARS	33	Moderate to Severe
10	II	M	16	36	140	6	ERT—9 YEARS	7	Mild to Moderate
11	II	M	23	47	129	18	ERT—5 YEARS	18	Moderate to Severe
12	II	M	12	31	124	2	ERT—8 YEARS	4	Severe
13	II	M	15	40	140	2	ERT—4 YEARS	5	Severe
14	II	M	10	41	125	4	ERT—6 YEARS	4	Severe
15	III	F	10	33	128	4	-	-	Mild to Moderate
16	III	M	5	20	107	1	-	-	Mild to Moderate
17	IV	M	13	28	99	1.5	ERT—6 YEARS	7	Moderate to Severe
18	IV	F	10	27	101	3	ERT—4 YEARS	6	Moderate to Severe
19	IV	F	13	17	99	2	ERT—6 YEARS	7	Moderate to Severe
20	VI	F	3	13	89	1	ERT—2 YEARS	1	Normal
21	VI	F	10	15	101	4	ERT—6 YEARS	4	Mild to Moderate
22	VI	M	9	20	110	2	ERT—6 YEARS	3	Moderate to Severe
23	VI	M	3	12	86	2	-	-	Normal
24	VI	M	17	31	105	0.7	ERT—10 YEARS	7	Severe
25	VI	M	6	14	88	6	-	-	Normal
26	VI	M	14	26	98	3	ERT—10 YEARS	4	Severe
27	VI	M	12	20	106	3	ERT—6 YEARS	6	Moderate to Severe
28	VI	F	9	17	107	0.5	ERT—8 YEARS	1	Moderate to Severe
29	VI	F	12	31	131	2	ERT—9 YEARS	3	Moderate to Severe
30	VI	M	7	19	108	1	ERT—5 YEARS	2	Normal

MPS, mucopolysaccharidosis; ERT enzyme replacement therapy; HSCT, hematopoietic stem cell tranplantation; LTBD, laryngotracheobronchial disease.

**Table 2 diagnostics-10-00037-t002:** Summary of the main endoscopic findings of laryngotracheobronchial disease according to MPS type.

MPS Type	Endoscopic Laryngeal Disease	Tracheobronchial Endoscopic Disease
MPS I (6 patients)	83.3% Supraglottic Redundancy—5 patients 50% Epiglottis thickening—3 patients 16.67% Tracheostomy—1 patient	50% with tracheal deformity—3 patients
MPS II (8 patients)	50% with laryngomalacia—4 patients 37.5% requiring tracheostomy—3 patients	75% with tracheobronchial disease—6 patients 25% Death in patients due to respiratory complications—2 patients
MPS III (2 patients)	100% Mild epiglottis thickening—2 patients 100% Mild laryngeal deformity—2 patients	Absence of tracheobronchial disease
MPS IV (3 patients)	Absence of laryngeal disease	100% Significant tracheal deformity—3 patients
MPS VI (11 patients)	Mild laryngeal disease 27.27% Laryngomalacia—3 patients	64% Tracheal deformity—7 patients

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
