# Peer review of "Laryngeal, Tracheal, and Bronchial Disease in the Mucopolysaccharidoses: Endoscopic Study"

_diagnostics, 2020, doi:10.3390/diagnostics10010037_

Round 1

Reviewer 1 Report

General aim of presented study was to describe laryngeal, tracheal and bronchial disease in the MPSs with the endoscopic study. Unfortunately for the Authors' assumption, examined group was not sufficient in number and too differentiated in the context of number of patients with specified MPS type, age, weight, duration of ERT, to draw significant conclusions.

Results section is presented rather poorly, lacking tables or graphs presenting acquired data. Table 1 (p6 and 7, v 204) should be completed with numbers and/or percentages of patients with selected endoscopic "findings" (even this term used in Table 1 description seems unproper for publication, also like "exam" in v.18, p1). In contrary in Discussion Authors thoroughly analysed results, but it's hard to follow.

Author Response

Dear Reviewer 1,

“General aim of presented study was to describe laryngeal, tracheal and bronchial disease in the MPSs with the endoscopic study. Unfortunately for the Authors' assumption, examined group was not sufficient in number and too differentiated in the context of number of patients with specified MPS type, age, weight, duration of ERT, to draw significant conclusions.”

Regarding your first opinion to the analyzed group, this is the number of patients included in the study; our hospital does not have more patients who could be incorporated into our sample. In any case we understand the criticism because this group is too small for absolute conclusions, which is addressed in our conclusions (v354, p11).

“Results section is presented rather poorly, lacking tables or graphs presenting acquired data. Table 1 (p6 and 7, v 204) should be completed with numbers and/or percentages of patients with selected endoscopic "findings" (even this term used in Table 1 description seems unproper for publication, also like "exam" in v.18, p1). In contrary in Discussion Authors thoroughly analysed results, but it's hard to follow.”

Table 1 was modified with the suggested numbers with their percentages and absolute numbers.

We exchanged the term "normal examination" to "normal airways" (v18, p1)

Reviewer 2 Report

Please, consider changes in:

Materials and Methods: When you propose your own LTBD severity score, please describe it in one paragraph – now it is dispersed in v. 116-125 and 136-146 plus Fig.1. As for Fig.1 – it is unclear and does not bring any additional information than provided in the text, in fact, in lacks mentioning TBD score as a component of LTBD score. You introduce the FB abbreviation for flexible bronchoscopy but use it only three times (v. 104, 110, and misspelled 348). In my opinion you might leave the full term. Results: Would it be possible to provide Patients characteristics in one table? I would much appreciate the patients to be individualized – so for all 30 patients you could give information in the following columns: MPS type, age, sex, weight, height, age at diagnosis, age at transplantation, age at ERT, snoring, sleep apnea, stridor, dyspnea, cyanosis (for the last five – would it be possible to tell how long they have been bothering patients who reported them?), LD score, TBD score, LTBD score. With this table, Fig.2. would be redundant (if you insist on keeping it, description on the y axis should be added). When you write “cyanosis episodes were reported in 9 patients (30%), two of them related to enzyme infusion and the remaining seven related to respiratory symptoms” (v. 163-164) what do you mean by the relation of cyanosis and enzyme infusion? Does the process of enzyme infusion cause cyanosis? In Table 1 for some manifestations you give percentage, for some not – how to read it? Perhaps when you give the full characteristics of all patients this table will not be needed anymore?

Author Response

Materials and Methods: When you propose your own LTBD severity score, please describe it in one paragraph – now it is dispersed in v. 116-125 and 136-146 plus Fig.1. As for Fig.1 – it is unclear and does not bring any additional information than provided in the text, in fact, in lacks mentioning TBD score as a component of LTBD score. You introduce the FB abbreviation for flexible bronchoscopy but use it only three times (v. 104, 110, and misspelled 348). In my opinion you might leave the full term. Results: Would it be possible to provide Patients characteristics in one table? I would much appreciate the patients to be individualized – so for all 30 patients you could give information in the following columns: MPS type, age, sex, weight, height, age at diagnosis, age at transplantation, age at ERT, snoring, sleep apnea, stridor, dyspnea, cyanosis (for the last five – would it be possible to tell how long they have been bothering patients who reported them?), LD score, TBD score, LTBD score. With this table, Fig.2. would be redundant (if you insist on keeping it, description on the y axis should be added). When you write “cyanosis episodes were reported in 9 patients (30%), two of them related to enzyme infusion and the remaining seven related to respiratory symptoms” (v. 163-164) what do you mean by the relation of cyanosis and enzyme infusion? Does the process of enzyme infusion cause cyanosis? In Table 1 for some manifestations you give percentage, for some not – how to read it? Perhaps when you give the full characteristics of all patients this table will not be needed anymore?

Dear Reviewer 2,

Regarding  your first observation we made a change in the table, making it clearer, showing that laryngotrachealbronchial disease DLTB) is the addition of laryngeal disease (LD) and tracheobronchial disease (TBD).

We replaced the acronym FB with flexible bronchoscopy.

We added a new table (Table 1) with the demographic data of the patients, but I thought it best to maintain figure 2, because it has the symptoms of patients and table 2 shows the endoscopic characteristics with the statistical data already modified.

Regarding cyanosis we did not find in the literature association with enzyme infusion, but we prefer to describe because it was a complaint of the families.

Reviewer 3 Report

The manuscript "Laryngeal, tracheal and bronchial disease in the Mucopolysaccharidoses: endoscopic study" is an original work, in a significant number of patients for a rare disease. However, and as the authors refer, the results should be taken with caution due to the heterogeneity of the series and the dispersion of cases. 

There are few publications in which the laryngoscopic examination of patients with MPS has been systematically collected. However, in this manuscript, in the methodology section it is not specified whether the exploration has been carried out in the context of a surgical intervention or if it has been a scheduled exploration to assess the disease involvement in which case informed consent is required by the parents or guardians and the approval for the corresponding ethical committee.

The authors do not specify and it is an important fact if the studied patients had previously undergone intubation because the laryngotracheal injuries associated to intubation appear in more than 40% of intubated pediatric patients, mainly caused by a defective technique and endotracheal tube pressure-induced mucosal damage

MPS I and MPS II patients in this study apparently have the most tracheal involvement and in this way will be established a set of recommendations for the intubation in these subtypes of MPS and a program of prevention of laryngotracheal lesions in these patients should be established.

Author Response

The manuscript "Laryngeal, tracheal and bronchial disease in the Mucopolysaccharidoses: endoscopic study" is an original work, in a significant number of patients for a rare disease. However, and as the authors refer, the results should be taken with caution due to the heterogeneity of the series and the dispersion of cases. 

There are few publications in which the laryngoscopic examination of patients with MPS has been systematically collected. However, in this manuscript, in the methodology section it is not specified whether the exploration has been carried out in the context of a surgical intervention or if it has been a scheduled exploration to assess the disease involvement in which case informed consent is required by the parents or guardians and the approval for the corresponding ethical committee.

The authors do not specify and it is an important fact if the studied patients had previously undergone intubation because the laryngotracheal injuries associated to intubation appear in more than 40% of intubated pediatric patients, mainly caused by a defective technique and endotracheal tube pressure-induced mucosal damage

MPS I and MPS II patients in this study apparently have the most tracheal involvement and in this way will be established a set of recommendations for the intubation in these subtypes of MPS and a program of prevention of laryngotracheal lesions in these patients should be established.

Dear Reviewer 3,

We added according to your guidance, in the material and methods section its two recommendations (v107-v108, p3): patients were evaluated to measure airway disease and none of them had previous intubation trauma.

The consent form is written on v130-v134, p3.

Round 2

Reviewer 1 Report

Dear Authors, after this minor revision I find this manuscript more clearly presenting acquired data and more valuable for publication.

Reviewer 2 Report

Thank you for taking my suggestions into consideration.

In the revised version I spotted one spelling mistake you might want to correct -  Table 1, heading of the 8th column - I believe you meant "treatment".

Reviewer 3 Report

The authors have made the suggested corrections, improving the understanding and quality of the work. It is acceptable to publish in this new version